# Chiral topographic instability in shrinking spheres

Fan Xu [1]✉, Yangchao Huang[1], Shichen Zhao[1] and Xi-Qiao Feng [2]✉

Many biological structures exhibit intriguing morphological patterns adapted to environmental cues, which contribute to their important biological functions and also inspire material designs. Here, we report a chiral wrinkling topography in shrinking core–shell spheres, as observed in excessively dehydrated passion fruit and experimentally demonstrated in silicon core–shells under air extraction. Upon shrinkage deformation, the surface initially buckles into a buckyball pattern (periodic hexagons and pentagons) and then transforms into a chiral mode. The neighbouring chiral cellular patterns can further interact with each other, resulting in secondary symmetry breaking and the formation of two types of topological network. We develop a core–shell model and derive a universal scaling law to understand the underlying morphoelastic mechanism and to effectively describe and predict such chiral symmetry breaking far beyond the critical instability threshold. Moreover, we show experimentally that the chiral characteristic adapted to local perturbation can be harnessed to effectively and stably grasp small-sized objects of various shapes and made of different stiff and soft materials. Our results not only reveal chiral instability topographies, providing fundamental insights into the surface morphogenesis of the deformed core–shell spheres that are ubiquitous in the real world, but also demonstrate potential applications of adaptive grasping based on delicate chiral localization.

Morphological pattern formation across length scales is energetically favourable for thin-walled living matter such as fruits[1,2], vegetables[3], leaves[4–6], embryos[7], organs[8], tumours[9] and brains[10], where spontaneous symmetry breaking during growth or dehydration is normally considered to be a crucial factor in their complex wrinkling topography[6,11,12]. For example, pollen grains of angiosperm flowers exhibit self-folding when exposed to a dry environment to prevent further desiccation[13]. Growth-induced residual stress accumulates during tumour progression, driving the global buckling collapse of blood and lymphatic vessels, which makes the vascular delivery of anticancer drugs ineffective[9]. Symmetry breaking in evolving wrinkling patterns during brain development results in the thickness difference between gyri and sulci, which is closely linked to neurodevelopment disorders such as lissencephaly, polymicrogyria, autism spectrum disorders and schizophrenia[14]. In terms of its practical use, symmetry breaking in the formation of surface morphology patterns has found ever-increasing applications in various fields, such as micro/nanofabrication of flexible electronic devices[15,16], surface self-cleaning and anti-fouling[17], synthetic camouflaging skins[18], shape-morphing soft actuators[19] and adaptive aerodynamic drag control[20]. The precise prediction, control and manipulation of reversible instability morphologies would be key for relevant applications.

Prior works[3,12,21–23] on morphological pattern formation in stressed spherical core–shells, a typical structure omnipresent in nature and industrial technologies, have demonstrated a variety of intriguing topographies such as dimple, buckyball and labyrinth modes. Here, we report a chiral instability topography in core–shell spheres. We observed that a drying passion fruit (*Passiflora edulia* Sims) initially buckles into a periodic buckyball pattern consisting of hexagons and

[1]Institute of Mechanics and Computational Engineering, Department of Aeronautics and Astronautics, Fudan University, Shanghai, P. R. China.
[2]Institute of Biomechanics and Medical Engineering, AML, Department of Engineering Mechanics, Tsinghua University, Beijing, P. R. China.
✉e-mail: fanxu@fudan.edu.cn; fengxq@tsinghua.edu.cn

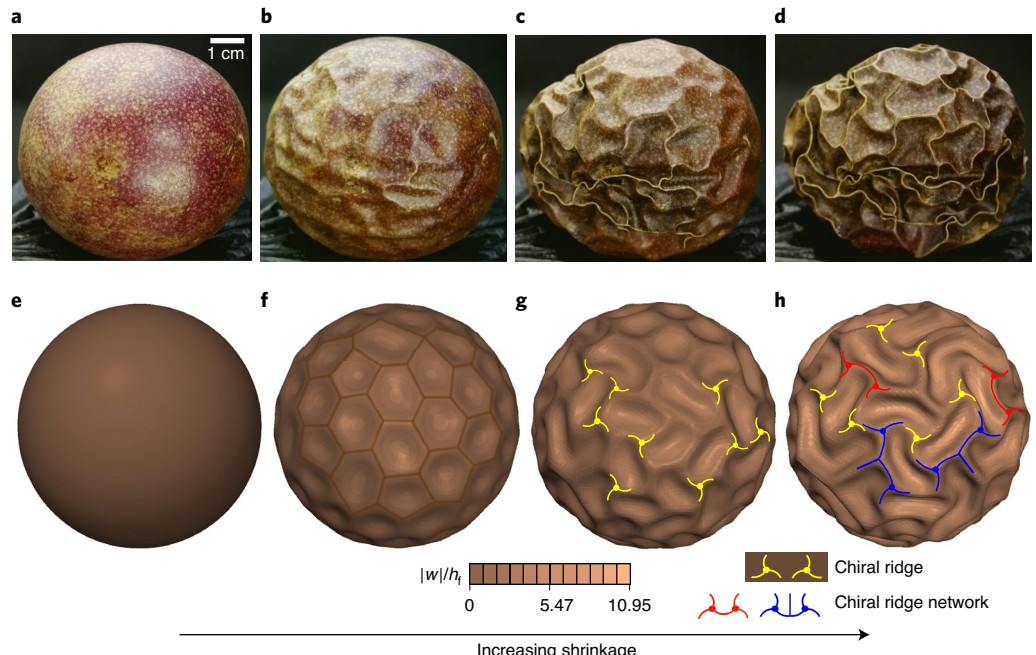

**Fig. 1 | Evolution of wrinkling topography in excessive dehydration of deformed passion fruit. a–h**, Natural observations (**a–d**) and model predictions (**e–h**) on day 1 (**a**,**e**), day 2 (**b**,**f**), day 4 (**c**,**g**) and day 7 (**d**,**h**). Upon shrinkage, the core–shell spheres first buckle into a buckyball pattern (periodic hexagons and pentagons in **b** and **f**) and then transform to a chiral ridge (**g**) and eventually to a ridge network (**h**) with the coalescence of neighbouring chiral ridges. The core experiences isotropic shrinking (Supplementary Sections I and II and Video 1).

pentagons, evolving into a chiral mode, and forms intriguing chiral topological networks upon excessive shrinkage (Fig. 1). Inspired by this natural phenomenon, we explored, both theoretically and experimentally, the morphological pattern formation and evolution of highly deformed core–shell spheres, especially the emergence of a chiral pattern and chiral ridge networks with symmetry breaking at the advanced bifurcation. We established a mathematical model and a scaling law to capture the chiral instability of core–shell spheres and explored a potential application of perturbation-adaptive chiral localization.

## Results

### Theory

To understand the underlying mechanism and to effectively predict the morphogenesis process, we consider an elastic spherical shell supported by a soft core. Upon shrinkage, the shell buckles elastically to relieve the compressive stress while the core concurrently deforms to maintain perfect bonding at the interface. In shallow shell theory[24], the coordinates of the core–shell system can be Cartesian in a tangent plane (or curvilinear and orthogonal). This framework can only describe a part of the spherical geometry (Extended Data Fig. 1), but it is competent here for theoretical analyses. The thickness of the surface layer is denoted by $h_f$, while the radius of the system is represented by $R$. The Young's modulus and Poisson's ratio of the surface layer are denoted by $E_f$ and $v_f$, respectively, while $E_s$ and $v_s$ are the corresponding material properties of the soft core. The elastic strain energy $\Pi_f$ in the shell can be written as the sum of the bending energy $\Pi_{ben}$ and membrane energy $\Pi_{mem}$ thus

$$
\begin{aligned}
\Pi_f &= \Pi_{ben} + \Pi_{mem} \\
&= \frac{1}{2} \iint_{\Omega_f} \left( D\mathbf{K}^T \bar{\mathbf{L}}_f \mathbf{K} + J\gamma^T \bar{\mathbf{L}}_f \gamma \right) dx\, dy \\
&= \frac{D}{2} \iint_{\Omega_f} \left( \kappa_x^2 + \kappa_y^2 + 2v_f\kappa_x\kappa_y + \frac{1-v_f}{2}\kappa_{xy}^2 \right) dx\, dy \\
&\quad + \frac{J_f}{2} \iint_{\Omega_f} \left( \gamma_x^2 + \gamma_y^2 + 2v_f\gamma_x\gamma_y + \frac{1-v_f}{2}\gamma_{xy}^2 \right) dx\, dy,
\end{aligned}
\tag{1}
$$

where $D = E_f h_f^3/[12(1 - v_f^2)]$ and $J_f = E_f h_f/(1 - v_f^2)$ stand for, respectively, the flexural and extensional rigidities of the shell, and $\bar{\mathbf{L}}_f$ represents the dimensionless elastic matrix. The membrane strain tensor and curvature tensor are denoted by $\gamma$ and $\mathbf{K}$, respectively. The elastic behaviour of the core can be described by a Winkler-type foundation[25,26] as

$$
\Pi_s = \frac{1}{2} \iint_{\Omega_s} K_s w^2\, dx\, dy,
\tag{2}
$$

in which $K_s = \bar{E}_s\sqrt{p^2 + q^2}/2R$ denotes the stiffness of the core[23,27], $w$ stands for deflection, $\bar{E}_s = E_s/(1 - v_s^2)$, and $p$ and $q$ represent the wavenumbers along the latitude and longitude directions, respectively.

The critical buckling of a core–shell sphere upon shrinkage is analogous to the hydrostatic instability of a spherical shell where an isotropic stress state remains in the pre-buckling stage, that is, $\sigma_{\alpha\beta}\delta_{\alpha\beta} = -\sigma$, in which $\delta_{\alpha\beta}$ is the Kronecker delta, $\sigma$ denotes the external hydrostatic pressure and the Greek indices $\alpha$ and $\beta$ take values in {1, 2}. According to Koiter's theory[24], elastic stability is primarily determined by the second variation of the total potential energy ($\Pi_t = \Pi_f + \Pi_s$), and one obtains the equilibrium partial differential equations by using the divergence theorem,

$$
\begin{aligned}
&u_{,xx} + \frac{1}{2}(1 - v_f)u_{,yy} + \frac{1}{2}(1 + v_f)v_{,xy} - \frac{1+v_f}{R}w_{,x} = 0, \\
&v_{,yy} + \frac{1}{2}(1 - v_f)v_{,xx} + \frac{1}{2}(1 + v_f)u_{,xy} - \frac{1+v_f}{R}w_{,y} = 0, \\
&D\nabla^4 w - \frac{J_f(1+v_f)}{R}\left(u_{,x} + v_{,y} - 2\frac{w}{R}\right) + \sigma h_f(w_{,xx} + w_{,yy}) \\
&\quad + K_s w = 0,
\end{aligned}
\tag{3}
$$

where a comma in a subscript denotes a partial derivative. As an ansatz, we consider the following forms for the displacements in the critical buckling state:

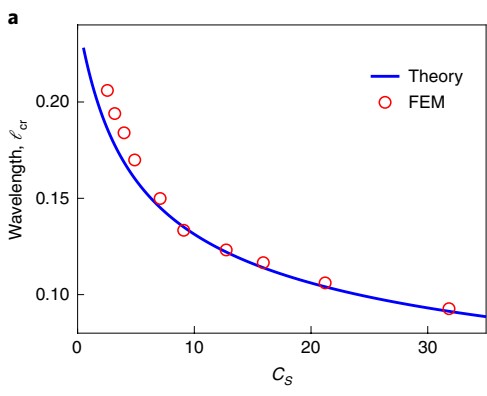

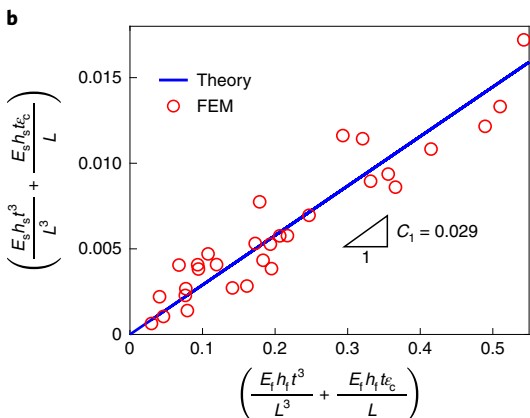

**Fig. 2 | Comparison of theoretical and numerical results. a**, The critical hexagonal wrinkling wavelength $\ell_{cr}$ as a function of the dimensionless parameter $C_s = (E_s/E_f)(R/h_f)^{3/2}$ that characterizes the modulus ratio and curvature. **b**, A scaling law (Methods) for the hexagonal-to-chiral mode transition. Our theoretical predictions agree well with FEM simulations, where $C_1$ denotes the slope.

$$u = A\sin(px/R)\cos(qy/R),$$
$$v = B\cos(px/R)\sin(qy/R), \qquad (4)$$
$$w = C\cos(px/R)\cos(qy/R),$$

in which $A$, $B$ and $C$ refer to the amplitudes of waves. Substituting equations (4) into equations (3) and minimizing with respect to $k = p^2 + q^2$, one obtains the critical conditions for the onset of wrinkling:

$$\frac{h_f^2}{4c^2R^2}k_{cr}^2 - \frac{\bar{E}_s R}{4E_f h_f}\sqrt{k_{cr}} - 1 = 0,$$

$$\frac{\sigma_{cr}}{E_f} = \frac{1}{k_{cr}} + \frac{h_f^2}{4c^2R^2}k_{cr} + \frac{K_s R^2}{E_f h_f k_{cr}}, \qquad (5)$$

$$\ell_{cr} = \frac{2\pi R}{\sqrt{k_{cr}}},$$

where $k_{cr}$, $\sigma_{cr}$ and $\ell_{cr}$ denote, respectively, the critical wavenumber, the compressive stress and the wavelength, $c = \sqrt{3(1 - v_f^2)}$. Here, we define a key dimensionless parameter $C_s = (E_s/E_f)(R/h_f)^{3/2}$ that characterizes the stiffness ratio of core–shells and the geometric curvature to classify pattern selection. Once the critical wavenumber $k_{cr}$ is solved, the theoretical buckling stress and wavelength can be calculated (Fig. 2a). During the natural dehydration process of passion fruit, the moduli of both the surface layer and the soft core may become larger (meaning that the surface layer and the core become stiffer), but we observed that the wrinkling wavelength in experiments (Fig. 1 and Supplementary Video 1) remains almost unchanged, and this critical wavelength $\ell_{cr}$ has some inherent (yet implicit) relation with the modulus ratio $E_s/E_f$ (equation (5)). Therefore, it is reasonable to approximate in the calculation that the modulus ratio $E_s/E_f$ remains relatively constant upon dehydration. Note that, although both natural and numerical observations (Fig. 1b,f) show that the buckyball pattern consisting of hexagons and pentagons covers the whole sphere (non-developable surface), the prevailing buckling mode in core–shell spheres is hexagonal. Also within the shallow shell framework (a part of sphere)[24], it is an analytical challenge to apply both hexagons and pentagons to describe the entire spherical surface. Hence, we assume this dominant hexagonal mode (displacement field) in equation (4), and the critical wrinkling condition based on our theory shows good agreement with numerical simulations. Equation (5), in fact, covers the classical buckling case of a spherical shell without a core ($K_s = 0$), for which there are explicit solutions for the critical threshold, that is, $\sigma_0 = E_f h_f/cR$, $k_0 = 2cR/h_f$ and $\ell_0 = \pi\sqrt{2Rh_f/c}$.

Although the critical buckling condition can be predicted analytically by using stability analysis, the secondary bifurcation with the hexagonal-to-chiral mode transition in the post-buckling stage remains a theoretical challenge. Here, we derived a scaling law to provide further insight into such chiral symmetry breaking far beyond the critical threshold (Methods). We assumed that each Y-shaped ridge in the wrinkling hexagons can be regarded as a bilayer system and thus that the chiral ridge instability of core–shell spheres can be simplified as the buckling of bilayered plates under compression. Minimization of the system energy leads to chiral strains that obey the linear relation in Fig. 2b, confirmed by numerical simulations.

### Computation

To trace the whole post-buckling topographic evolution, we applied the finite element method (FEM) by accounting for various geometric and material parameters (Supplementary Section II). The main challenge lies in the solution of nonlinear equations, since multiple solution branches in the post-buckling regime can be connected via multiple bifurcations. Moreover, for instabilities that are extremely localized (for example, the ridge network shown in Fig. 1c,d), there must exist a local transfer of elastic strain energy from one part of the system to the neighbouring regions, and global solution methods may encounter difficulties in convergence. To solve this difficulty, we implemented a pseudodynamic algorithm by introducing velocity-dependent damping and inertial terms, which can be naturally viewed as a perturbation to allow the calculation to pass through the unstable transitions and to trigger chiral symmetry breaking (Methods). The bifurcation portraits of the dimensionless deflection $|w|/h_f$ for various core–shell spheres with different $C_s$ upon shrinkage are plotted in Fig. 3. Periodic buckyball (with hexagons prevailing) wrinkling patterns with supercritical bifurcation emerge initially at the critical thresholds. Upon further shrinkage, hexagonal-to-chiral mode transitions occur, where Y-shaped ridges in the wrinkling hexagons may buckle into chiral ridges. Neighbouring chiral cellular modes can further interact with each other to form two types of topological network. While symmetry is eventually broken with further shrinkage, leading to universal hexagonal-to-chiral mode transitions, different $C_s$ values result in different critical thresholds and wavelengths for the buckyball (with hexagon dominating) buckling mode.

### Experiment

Guided by this theoretical understanding, we next designed a demonstrative experiment to harness such an instability mechanism to achieve pattern tunability, by using liquid silicone that can solidify into any desired shape in a well-designed mould. We made a

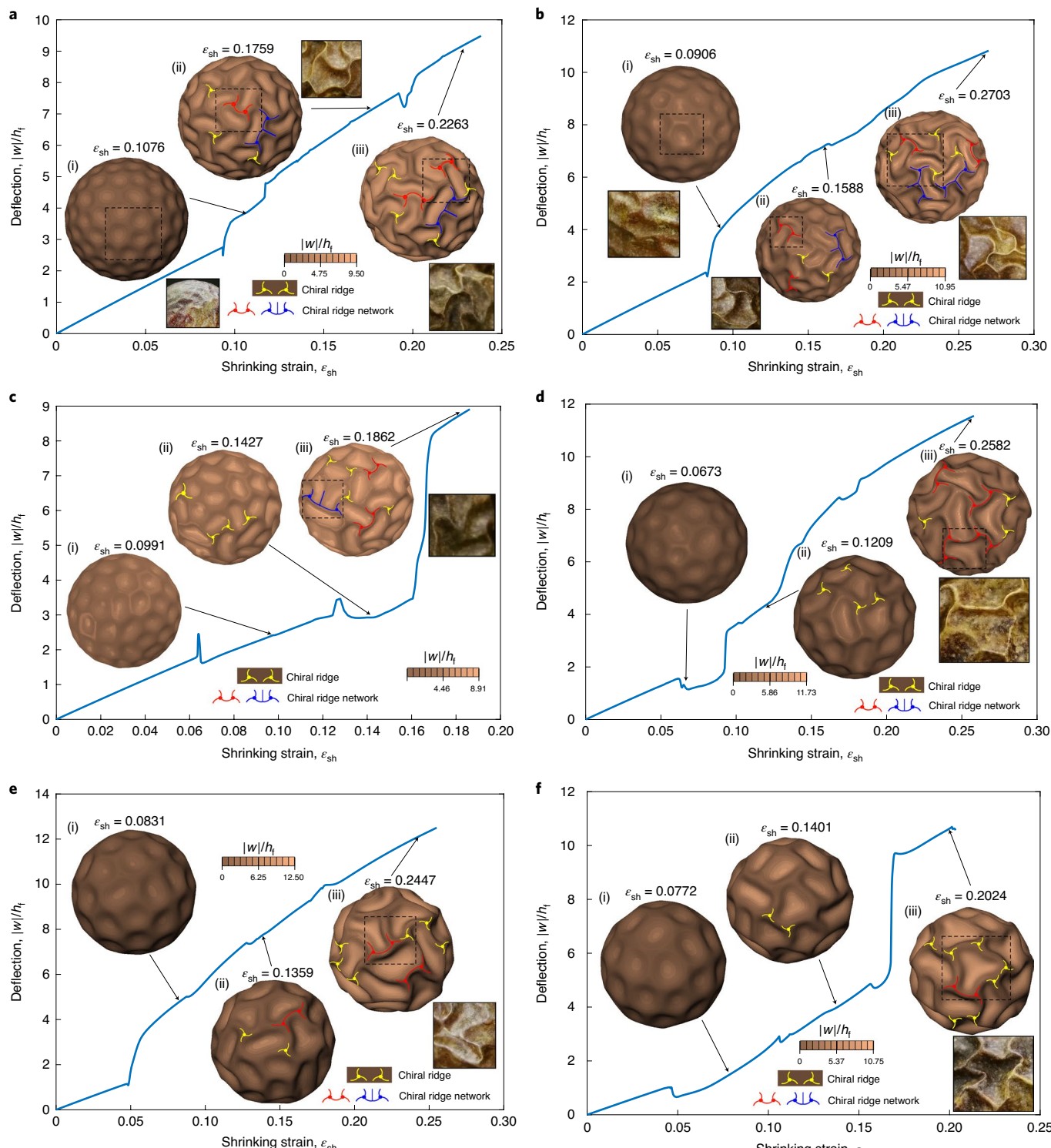

**Fig. 3 | Bifurcation diagrams of post-buckling morphology evolutions in core–shell spheres with different $C_s$ upon shrinkage. a–f**, Diagrams for $C_s$ values of 12.7 (**a**), 9.09 (**b**), 7.07 (**c**), 3.98 (**d**), 3.18 (**e**) and 2.55 (**f**), showing the buckyball pattern (with hexagons prevailing) (i) and chiral ridge networks (ii and iii). Excess shrinkage leads to advanced symmetry breaking of the buckyball mode, transforming into the chiral mode and the chiral ridge network eventually.

spherical shell with a hexagonal pattern on the surface, a cavity and a small hole (diameter ~4 mm) for air extraction to induce shrinkage (Methods). Since silicone has a much lower elastic modulus than passion fruit, the smooth shell structure does not buckle into hexagonal patterns (cannot reach the advanced bifurcation range shown in Fig. 3) but exhibits global deformation upon pressure loading condition by air extraction (Methods and Supplementary Video 5).

To focus on the chiral bifurcation and to facilitate instability morphology control at this bifurcation, we fabricated artificial hexagonal patterns on the shell surface. We extracted air slowly (~2 mL s⁻¹) from the sample to control the pressure (~10 kPa) so that a state of homogeneous compression could be perfectly achieved. Notably, these well-designed hexagonal networks on the surface of the sample buckle into chiral patterns (Fig. 4a–d and Supplementary Video 2),

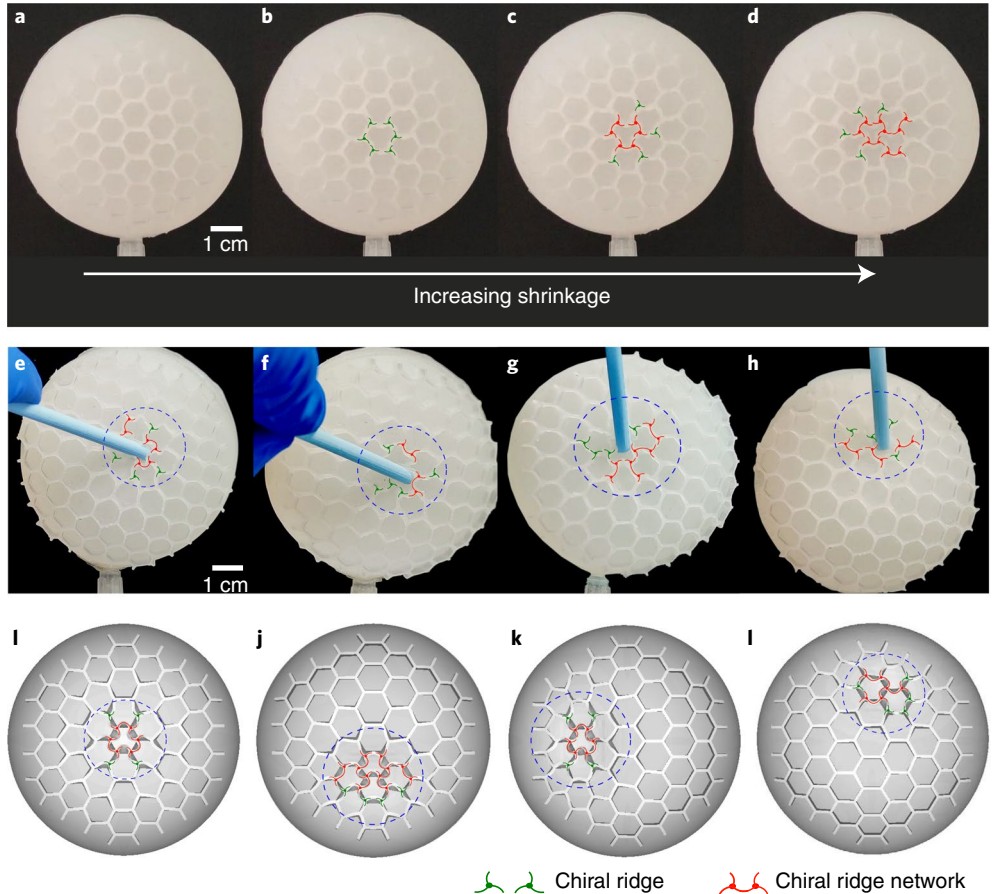

Chiral ridge          Chiral ridge network

**Fig. 4 | Air extraction-induced chiral topography on curved surfaces.**
**a–d**, The experimental formation of a chiral ridge network with continuous air extraction, showing the hexagonal-to-chiral mode transition with increasing shrinkage of core–shells (Supplementary Video 2). **e–l**, The localization of tunable chiral networks on curved surfaces (Supplementary Video 3) triggered by a perturbation (poke by a rod) in experiments (**e–h**), consistent with numerical simulations (**i–l**).

analogous to the observation of highly dehydrated passion fruits and model predictions (Fig. 1). Furthermore, we can flexibly control the position of local chiral networks by imposing external perturbation as illustrated in Fig. 4e–h (Methods and Supplementary Video 3), consistent with FEM simulations in Fig. 4i–l. These experiments not only demonstrate a hexagonal-to-chiral mode transition, consistent with our theoretical predictions, but also shed light on rational designs of controllable chiral patterns.

### Adaptive grasping
Based on these insights, we show that this perturbation-induced chiral instability can be harnessed to effectively and stably grasp small-sized objects with different geometries and made of different stiff or soft materials. The object to grasp acts as a local perturbation when in contact with the hexagonal-patterned shell and is then adaptively locked by the induced local chiral networks. Similar to the aforementioned experimental setup, we fabricated a hemispherical shell with a hexagonal surface pattern as the main body of the gripper. A small hole was made at the bottom of the cap for air extraction. Then, the whole gripper was fixed onto a lifting frame to steadily control the movement. When the curved hemispherical cap touches the target, the contact perturbation-induced symmetry breaking triggers chiral network localization. The chiral pattern and the interface friction spontaneously adapt to the interactions at the contacting areas, which are naturally influenced by the shape and stiffness of the object, so that different objects can be grasped by this smart locking together with air extraction (Fig. 5, Supplementary

Fig. 4 and Video 4). When we restored the pressure difference, that is, inflated the cap cavity, the chiral networks elastically reverted back to hexagons, releasing the grasped object. The contrast experiments showed that the hemispherical caps with a smooth surface (no chiral instability) could not grasp those objects at all (Supplementary Video 5), supporting the critical role of the chiral network localization in the grasping process.

### Discussion
We have unveiled chiral-mode symmetry breaking during excessive shrinkage of core–shell spheres, which can be formulaically described and precisely predicted by our theories and computations, in good agreement with carefully designed experiments. Beyond the critical buckyball wrinkling, chiral ridges emerge on the curved surfaces upon excess deformation, and the neighbouring chiral cellular Y-shaped modes can further interact with each other to form advanced chiral topological networks. The critical buckyball wrinkling conditions can be obtained analytically by using linear stability analysis, while strong nonlinearity (both geometric and material) in the post-buckling regime of shrinking spheres results in considerable difficulties in the theoretical predictions of advanced bifurcations and their associated morphological patterns. Consequently, theoretical analyses on secondary and multiple bifurcations of chiral instability have to resort to dimensional analysis (scaling law) based on certain simplified models. From the computational standpoint, the major challenge in extremely shrinking spheres at large strain is the solution of highly nonlinear equations. The most classical solution method to solve nonlinear static

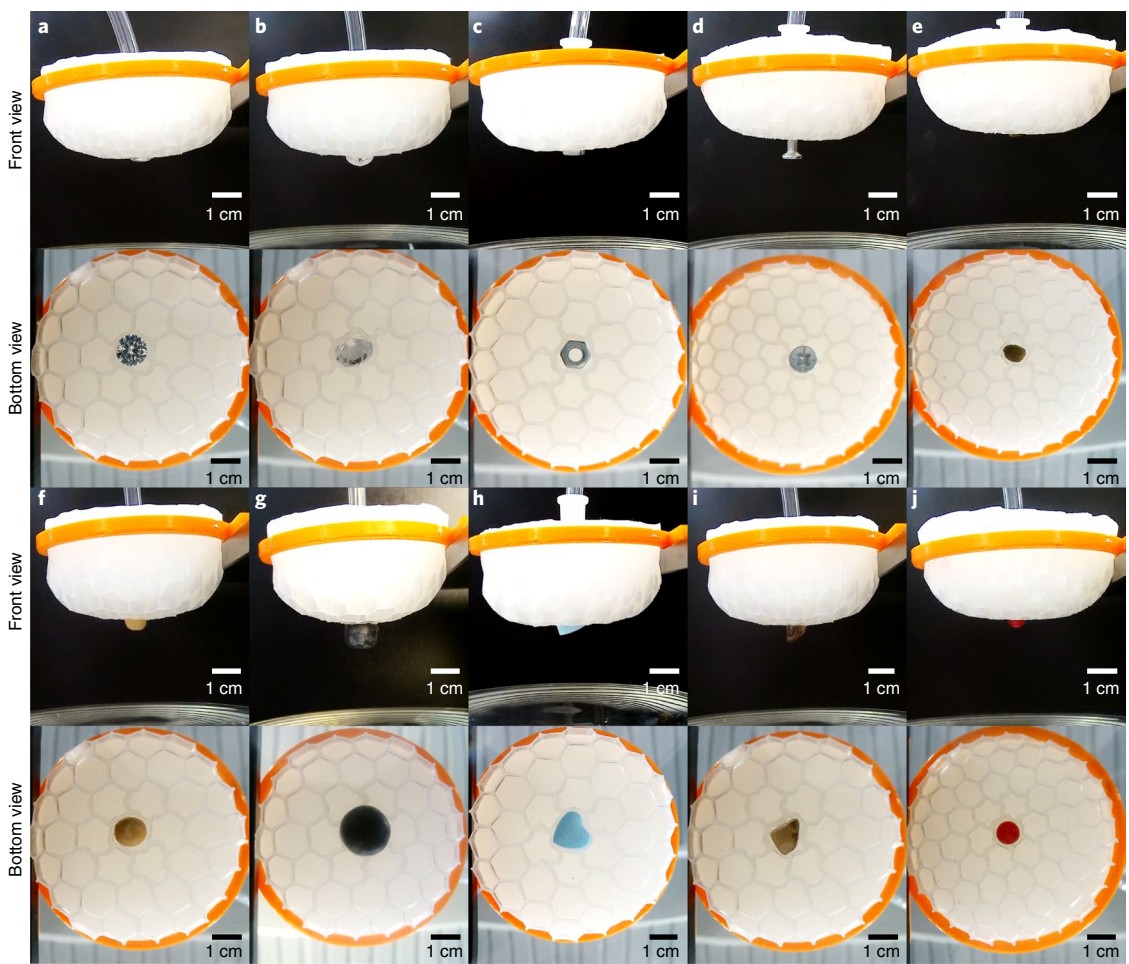

**Fig. 5 | Topographic grasping experiments on objects of different geometry, size and material. a–j**, Grasping of different objects: diamond (**a**,**b**), nut (**c**), screw (**d**), mung bean (**e**), soya bean (**f**), blueberry (**g**), heart-shaped candy (**h**), irregular shaped glass (**i**) and glass ball (**j**). The chiral deformation enables effective, target-adaptive grasping (Supplementary Video 4).

problems is the path-following continuation technique such as that of Riks, while numerical convergence cannot always be ensured for extreme wrinkling problems upon large deformations, since a large number of solution branches can be connected via multiple bifurcations. This fact motivated us to apply the dynamic relaxation method to leap over some localized energy barriers in the nonlinear evolution paths, while the dynamic method cannot straightforwardly predict subcritical bifurcations and hysteresis. Making progress in both theoretical and computational analyses of multiple bifurcations in highly nonlinear evolution paths might require more advanced mathematical approaches.

Inspired by the chiral instability topography induced by local perturbation, we demonstrated an exemplar application of target-adaptive grasping based on chiral localization, while future work may take advantage of smart active materials such as hard-magnetic soft materials and liquid-crystal elastomers to enhance multifunctional designs under multiphysics stimuli. Our results not only provide physical insights into the wrinkling topography of highly deformed core–shell spheres by a universal law but also pave a promising way for realizing multifunctional surfaces by harnessing fruitful topography on curved geometry.

## Methods
### Dimensional analysis of chiral instability
We carried out dimensional analysis to predict the chiral bifurcation of core–shell spheres (Extended Data Fig. 1) upon dehydration

(equivalent to thermal shrinkage). Based on the experimental observations and numerical calculations, we assumed that each cellular ridge before chiral instability can be viewed as a layered plate and thus the chiral bifurcation of a cellular ridge can be simplified as the buckling of a bilayer subject to shrinking strain (Extended Data Fig. 1c). Such a plate-like ridge has length $L$ and thickness $t$ and comprises an upper layer of width $h_f$ and a lower layer of width $h_s$. Each layer has a Young's modulus $E_\zeta$, Poisson's ratio $v_\zeta$ and bending stiffness $D_\zeta = E_\zeta t^3/[12(1 - v_\zeta^2)]$, where $\zeta$ is 'f' or 's'.

The bending energies of the upper and lower layers can be expressed as

$$\mathcal{P}_f^b = \frac{D_f}{2} \iint_{\Omega_1} \left[ \left(u_{f,zz} + u_{f,yy}\right)^2 + 2\left(1 - v_f\right)\left(u_{f,yz}^2 - u_{f,zz}u_{f,yy}\right) \right] d\Omega_1, \quad (6)$$

$$\mathcal{P}_s^b = \frac{D_s}{2} \iint_{\Omega_2} \left[ \left(u_{s,zz} + u_{s,yy}\right)^2 + 2\left(1 - v_s\right)\left(u_{s,yz}^2 - u_{s,zz}u_{s,yy}\right) \right] d\Omega_2, \quad (7)$$

where $u_f$ and $u_s$ denote, respectively, the out-of-plane deflection of the upper and lower layers, while $\Omega_1$ and $\Omega_2$ represent the area of the mid surface of the upper and lower layer, respectively.

As an ansatz, we consider the following forms for the deflections in the chiral buckling state:

$$u_f = \Phi_f(z) \sin \frac{\pi}{L} y, \quad (8)$$

$$u_s = \Phi_s(z) \sin\frac{\pi}{L}y, \tag{9}$$

where the functions $\Phi_f(z)$ and $\Phi_s(z)$ can be expanded into series of exponential decay functions as

$$\Phi_f(z) = \sum_i A_{fi}(k_{fi}z), \tag{10}$$

$$\Phi_s(z) = \sum_i A_{si}(k_{si}z), \tag{11}$$

where $k_{fi}$ and $k_{si}$ are coefficients of the following order:

$$k_{fi} \sim k_{si} \sim \frac{1}{L}, \tag{12}$$

and the displacement continuity condition is satisfied at the interface of upper and lower layers, that is, $\Phi_f(h_s) = \Phi_s(h_s)$.

According to equations (8) to (12), one obtains

$$u_{,zz} \sim u_{,yy} \sim u_{,yz}. \tag{13}$$

Substituting equation (13) into equations (6) and (7), the bending energies read

$$\mathcal{P}_f^b \sim \frac{E_f t^3}{L^4} \iint_{\Omega_1} \left[\sum_i A_{fi}(k_{fi}z)\sin\left(\frac{\pi y}{L}\right)\right]^2 dy\,dz \sim \frac{E_f t^3 h_f}{L^3}a_1, \tag{14}$$

$$\mathcal{P}_s^b \sim \frac{E_s t^3}{L^4} \iint_{\Omega_2} \left[\sum_i A_{si}(k_{si}z)\sin\left(\frac{\pi y}{L}\right)\right]^2 dy\,dz \sim \frac{E_s t^3 h_s}{L^3}a_2, \tag{15}$$

in which $a_1 = \iint \left[\sum_i A_{fi}(k_{fi}\tilde{z}h_f)\sin(\pi\tilde{y})\right]^2 d\tilde{y}\,d\tilde{z}$,

$a_2 = \iint \left[\sum_i A_{si}(k_{si}\tilde{z}h_s)\sin(\pi\tilde{y})\right]^2 d\tilde{y}\,d\tilde{z}$, $\tilde{y} = y/L$ and $\tilde{z} = z/h_\zeta$.

The membrane energy can be determined by the in-plane strains given by (note that, for simplicity, the subscript $\zeta$ has been omitted)

$$\varepsilon_{yy}^0 = \frac{\partial v}{\partial y} + \frac{1}{2}\left(\frac{\partial u}{\partial y}\right)^2 + \varepsilon_{sh}, \tag{16}$$

$$\varepsilon_{zz}^0 = \frac{\partial w}{\partial z} + \frac{1}{2}\left(\frac{\partial u}{\partial z}\right)^2 + \varepsilon_{sh}, \tag{17}$$

$$\varepsilon_{yz}^0 = 0, \tag{18}$$

where $\varepsilon_{sh}$ is the thermal shrinking strain, and $v$ and $w$ represent the in-plane displacements in the mid surface along the $y$ and $z$ directions, respectively, the order of which can be determined by minimizing the membrane energy. Consequently, the in-plane displacements in the mid surface can be approximated as $v = By$ and $w = Cz$, in which $B$ and $C$ refer to the slopes of variation.

The membrane energies of the upper and lower layers can be expressed as

$$\mathcal{P}_f^m = \frac{E_f t}{2(1-v_f^2)} \iint_{\Omega_1} \left\{\left[(\varepsilon_{yy}^0)_f + (\varepsilon_{zz}^0)_f\right]^2 + 2(1-v_f) \left[(\varepsilon_{yz}^0)_f^2 + (\varepsilon_{yy}^0)_f(\varepsilon_{zz}^0)_f\right]\right\} d\Omega_1, \tag{19}$$

$$\mathcal{P}_s^m = \frac{E_s t}{2(1-v_s^2)} \iint_{\Omega_2} \left\{\left[(\varepsilon_{yy}^0)_s + (\varepsilon_{zz}^0)_s\right]^2 + 2(1-v_s) \left[(\varepsilon_{yz}^0)_s^2 + (\varepsilon_{yy}^0)_s(\varepsilon_{zz}^0)_s\right]\right\} d\Omega_2. \tag{20}$$

According to equations (8) to (12) and (16) to (18), the membrane energies read

$$\mathcal{P}_f^m \sim \frac{E_f t \varepsilon_{sh}}{L^2} \iint_{\Omega_1} \left[\sum_i A_{fi}(k_{fi}z)\sin\left(\frac{\pi y}{L}\right)\right]^2 dy\,dz \sim \frac{E_f h_f t \varepsilon_{sh}}{L}a_1, \tag{21}$$

$$\mathcal{P}_s^m \sim \frac{E_s t \varepsilon_{sh}}{L^2} \iint_{\Omega_2} \left[\sum_i A_{si}(k_{si}z)\sin\left(\frac{\pi y}{L}\right)\right]^2 dy\,dz \sim \frac{E_s h_s t \varepsilon_{sh}}{L}a_2. \tag{22}$$

Since the upper and lower layers buckle simultaneously, combining equations (14), (15), (21) and (22) leads to

$$\mathcal{P}_f^b + \mathcal{P}_f^m \sim \mathcal{P}_s^b + \mathcal{P}_s^m, \tag{23}$$

namely,

$$\left(\frac{E_f h_f t^3}{L^3} + \frac{E_f h_f t \varepsilon_{sh}}{L}\right)a_1 \sim \left(\frac{E_s h_s t^3}{L^3} + \frac{E_s h_s t \varepsilon_{sh}}{L}\right)a_2. \tag{24}$$

Note that $a_1/a_2$ is a non-negative constant. Based on calculations and equation (24), the scaling law yields the following explicit form for the chiral shrinking strain $\varepsilon_c$:

$$C_1\left(\frac{E_f h_f t^3}{L^3} + \frac{E_f h_f t \varepsilon_c}{L}\right) = \left(\frac{E_s h_s t^3}{L^3} + \frac{E_s h_s t \varepsilon_c}{L}\right), \tag{25}$$

where $C_1 = 0.029$ is a fitting coefficient. The scaling law in equation (25) agrees well with finite element simulations for chiral bifurcation (Fig. 2b).

### Numerical method

We performed finite element simulations in commercial software Abaqus based on parameters similar to experimental observations. Since the deformation of core–shell spheres can be large (up to 30% shrinking strain), we applied the widely used hyperelastic neo-Hookean (nHk) constitutive law for both the surface layer and the soft core, while more sophisticated hyperelastic constitutions such as the Mooney–Rivlin (MR) model were also examined but showed trivial quantitative differences that did not change the substantial nonlinear mechanism of the instability problem. The elastic strain energy density function of the nHk model is defined as

$$\Psi_{nHk} = C_{10}(I_1 - 3) + \frac{1}{D_1}(J-1)^2, \tag{26}$$

in which $C_{10} = E/4(1+v)$ and $D_1 = 6(1-2v)/E$ are material parameters. The volume change reads $J = \det(\mathbf{F})$, where $\mathbf{F}$ is the deformation gradient tensor. The first strain invariant reads $I_1 = \mathrm{tr}(\mathbf{F}^T \cdot \mathbf{F})$. We coupled eight-node hexahedral volume (C3D8R) elements for the soft core and thin shell (S4R) elements for the surface layer by using a 'tie' constraint at the interface. Mesh convergence was carefully examined for all simulations. The main challenge is the solution of the nonlinear equations, as numerous post-buckling solution branches can be connected via multiple bifurcations[23,28]. Therefore, we applied the dynamic relaxation method to allow the calculation to pass through the unstable transitions, which introduces velocity-dependent damping ($\mathbf{C}$) and artificial inertial ($\mathbf{M}$) terms into the static equilibrium equation ($\mathbf{R}(\mathbf{U}, \lambda) = 0$), leading to

$$\mathbf{M}\frac{d^2\mathbf{U}}{dt^2} + \mathbf{C}\frac{d\mathbf{U}}{dt} + \mathbf{R}(\mathbf{U}, \lambda) = 0, \tag{27}$$

where $\mathbf{R}$ is the residual force, $\mathbf{U}$ denotes unknown variables and $\lambda$ represents an incremental loading parameter. Realistic definitions of mass and damping were not necessary; thus, we set these quantities to obtain optimal convergence of $t \to \mathbf{U}(t)$ for large values of time $t$ (no

physical meaning here). When the model is stable (quasi-static), viscous energy dissipation remains quite small such that the artificial damping does not notably perturb the solution. When the system tends to be dynamically unstable, nodal velocities increase, and thus, part of the elastic strain energy released can be dissipated by the damping. A shrinkage load (equivalent to thermal expansion or residual strain) was applied to the core while the surface layer was loading free, which can be expressed as

$$\varepsilon_{\text{sh}} = \alpha \Delta T \mathbf{I} \quad \text{with} \quad \Delta T < 0, \tag{28}$$

where $\alpha$, $\Delta T$ and $\mathbf{I}$ stand for the thermal expansion coefficient, temperature change and second-order identity tensor, respectively. The shrinkage load $\varepsilon_{\text{sh}}$ can also be characterized by an isotropic residual strain $\varepsilon_{\text{sh}} = \varepsilon_{\text{res}} = -\lambda \mathbf{I}$. In the numerical calculations shown in Fig. 1e–h, we took $R/h = 50$ and $C_{\text{s}} = (E_{\text{s}}/E_{\text{f}})(R/h_{\text{f}})^{3/2} = 9.09$.

## Experimental method for realizing functional chiral surfaces

To realize flexible tunability of chiral patterns and to further harness the hexagonal-to-chiral mode transition for achieving smart surfaces, we designed demonstrative experiments based on air extraction from silicon core–shell spheres. The simple experimental system consists of two combined hemispherical caps with a channel connecting the internal cavity and an external tube for air extraction. To achieve a hexagonal network on the surface of the hemispherical cap, we designed a mould with a hexagonal network by applying three-dimensional printing technology. Then, we poured in two-part liquid silicone (Hongyejie Technology Co. Ltd.) in 1:1 mass ratio. Liquid silicone needs to stand for 3 hours at 25 °C to cure fully. To create a cavity in the centre of the sample, we applied a hemispherical lid with a diameter slightly smaller than the outer diameter to cover the bottom of the mould when the liquid silicone was curing. After the liquid silicone had cured and was demoulded, we glued two identical hemispherical caps together. The typical parameters of the samples were an outer diameter of $2R = 70$ mm, a diameter of the inner cavity of $2r = 58$ mm and a hexagonal cellular length of $L = 4.33$ mm, height of $H = 2.61$ mm and thickness of $t = 0.75$ mm. The experimental procedure to realize functional chiral surfaces is illustrated in Extended Data Fig. 2. The inner cavity of the samples was pumped out and depressurized to create a state of homogeneous shrinkage. To demonstrate the effects of shrinkage on the hexagonal-to-chiral mode transition, we slowly exhausted the air in the samples to mimic dehydration-induced shrinkage of passion fruit. When the samples deformed elastically to certain values, the hexagonal network lost stability and buckled into a chiral topography (Fig. 4a–d). Note that this mode transition is reversible when the air re-enters the sample and the pressure difference is restored. To further illustrate the tunability of the chiral localization, we applied a small disturbance (poke by a rod) somewhere on the surface to trigger the hexagonal-to-chiral mode transformation (Fig. 4e–h) while the sample was subjected to homogeneous shrinkage, which was in good agreement with finite element simulations (Fig. 4i–l). This strategy can provide enlightenment for the design of programmable functional surfaces such as adaptive grasping based on chiral localization.

## Chiral topography for adaptive grasping

Based on the aforementioned experiment, we present a target-adaptive gripper which can grasp small objects based on a hexagonal-to-chiral mode transformation. Simple structure, easy control, shape adaptation and filterable grasping are prominent advantages of the chiral gripper. The gripper system consists of a hemispherical shell with hexagonal topography, an air channel and a lifting frame that can move up and down (Supplementary Fig. 3). The air channel and the hemispherical part constitute a cavity structure, the former being connected to an external exhaust device to trigger the hexagonal-to-chiral mode transition by air extraction. The lifting frame is combined with the cap to

control the motion. The working principle of the gripper is introduced as follows: The lifting frame descends to make the gripper approach a target. When the hexagonal network on the curved surface touches the object, the contact perturbation triggers the hexagonal-to-chiral topographic deformation that can well fit with the targeted shape. Then, the exhaust device begins to pump air. With increasing air extraction, the chiral topography can lock the object tightly to achieve a stable grasp. Finally, the object leaves the desk when raising the lifting frame. When the pressure difference is restored, the chiral topography elastically reverts back to hexagonal networks, releasing the grasped object. We carried out topographic grasping experiments on stiff or soft objects of different shapes and sizes (Fig. 5 and Supplementary Fig. 4). Our experiments showed that the gripper can smartly and stably grasp various small-sized objects. To further demonstrate the crucial role played by the chiral topography in robust grasping, we performed contrast experiments by making a hemispherical cap with a smooth surface. Except for the lack of the initial hexagonal network on the surface, the other parameters of the gripper remained exactly the same as in the aforementioned grasping experiments. With the smooth surface, the targets slid off, leading to failure of effective grasping (Supplementary Video 5). Our experiments not only prove the critical role of the chiral topography in effective, target-adaptive grasping but also shed light on smart gripper designs.

## Reporting summary

Further information on research design is available in the Nature Research Reporting Summary linked to this article.

## Data availability

Source data for the FEM computations shown in Figs. 2 and 3 are available with this manuscript.

## Code availability

The code used in this study can be obtained from Zenodo[29].

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

## Acknowledgements

This work is supported by the National Natural Science Foundation of China (grants no. 12122204, 11872150 and 11921002), Shanghai Pilot Program for Basic Research-Fudan University (grant no. 21TQ1400100-21TQ010), Shanghai Shuguang Program (grant no. 21SG05), Shanghai Rising-Star Program (grant no. 19QA1400500) and young scientist project of the MOE innovation platform.

## Author contributions

F.X. and X.-Q.F. conceived the idea. F.X. designed the research. Y.H. and S.Z. conducted the experiments. F.X. and Y.H. developed the theoretical models and carried out the dimensional analyses. Y.H. and S.Z. performed the numerical simulations. F.X., Y.H. and S.Z. interpreted the results. F.X. and Y.H. wrote the manuscript. All the authors provided helpful discussions.

## Competing interests

The authors declare that they have no competing interests.

## Additional information

**Extended data** is available for this paper at https://doi.org/10.1038/s43588-022-00332-y.

**Correspondence and requests for materials** should be addressed to Fan Xu or Xi-Qiao Feng.

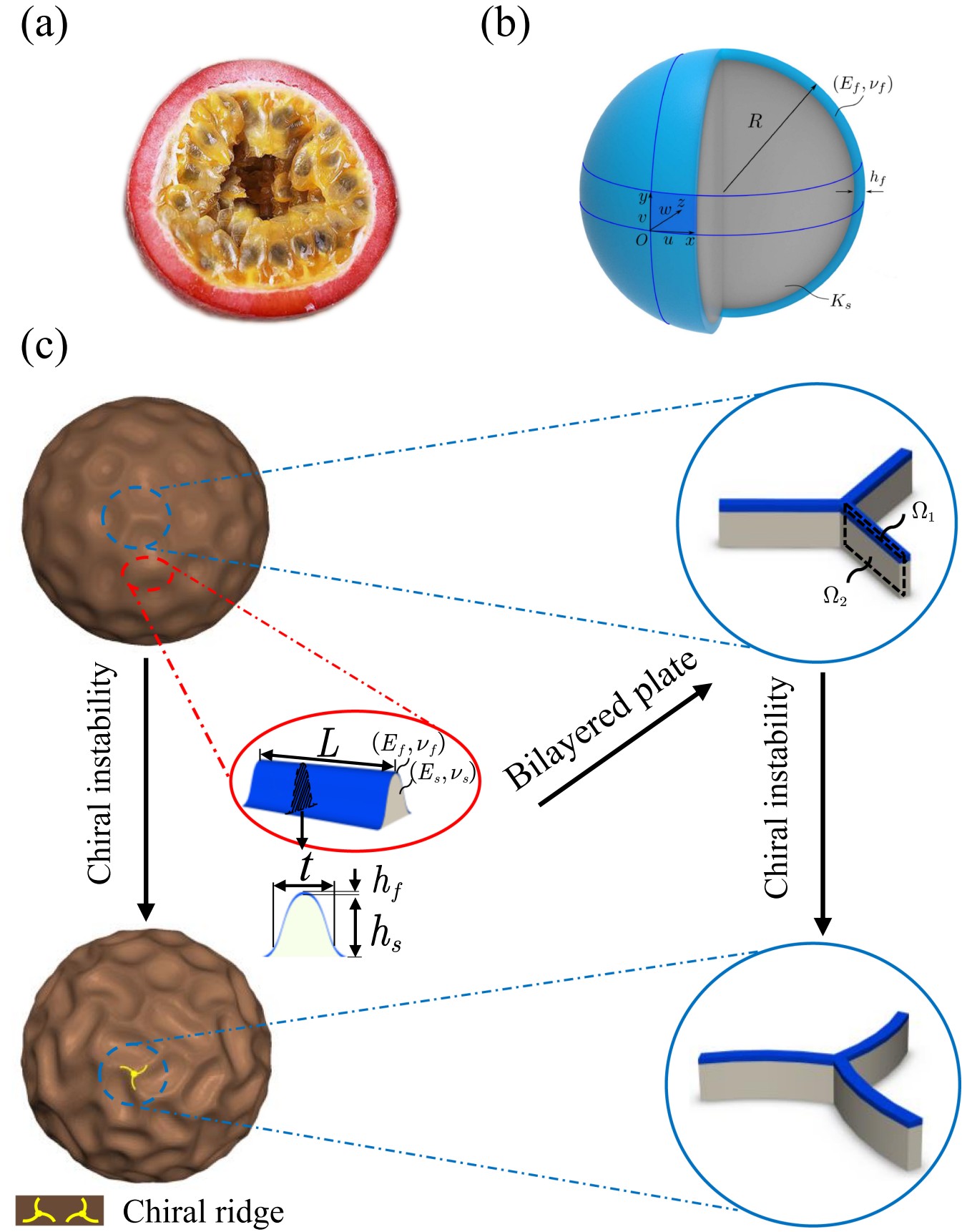

**Extended Data Fig. 1 | Passion fruit (Passiflora edulia Sims).** (a) Cross section. (b) Geometry of a core-shell sphere. (c) Schematic of chiral buckling of a Y-shaped cellular representative layered plate.

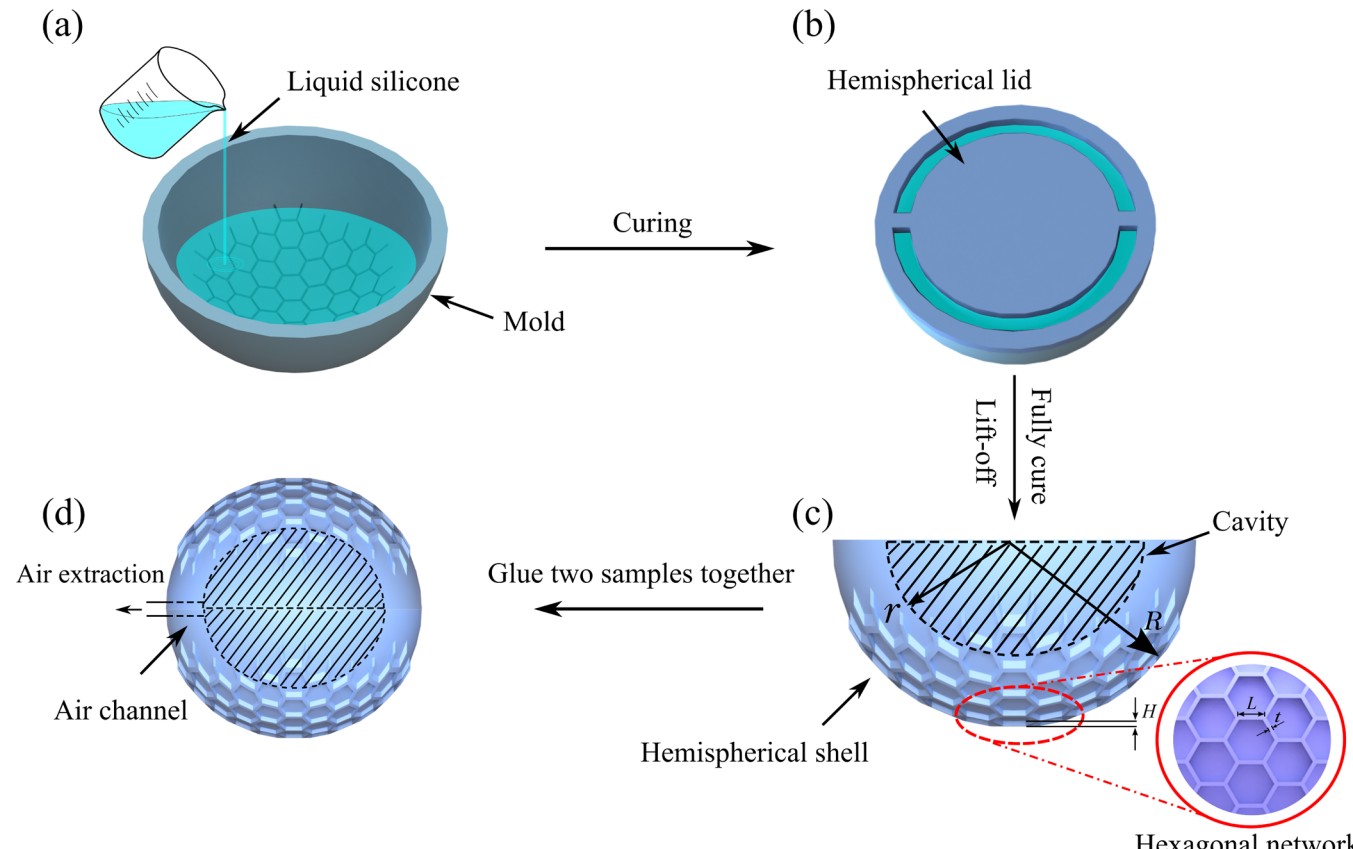

**Extended Data Fig. 2 | Experimental process of realizing functional chiral topography.** (a) Pour liquid silicone on a 3D printed mold with hexagonal network on the surface. (b) Create a cavity in the sample by using a hemispherical cover when the liquid silicone is curing. (c) A silicon hemispherical shell with hexagonal pattern on the surface. (d) Two hemispherical shells with hexagonal network are glued and can be separated, with a channel connecting the internal cavity and external tube for air extraction.

# nature research

# Reporting Summary

Nature Research wishes to improve the reproducibility of the work that we publish. This form provides structure for consistency and transparency in reporting. For further information on Nature Research policies, see our Editorial Policies and the Editorial Policy Checklist.

## Statistics

For all statistical analyses, confirm that the following items are present in the figure legend, table legend, main text, or Methods section.

| n/a | Confirmed | |
|---|---|---|
| ☐ | ☒ | The exact sample size (*n*) for each experimental group/condition, given as a discrete number and unit of measurement |
| ☐ | ☒ | A statement on whether measurements were taken from distinct samples or whether the same sample was measured repeatedly |
| ☒ | ☐ | The statistical test(s) used AND whether they are one- or two-sided *Only common tests should be described solely by name; describe more complex techniques in the Methods section.* |
| ☐ | ☒ | A description of all covariates tested |
| ☐ | ☒ | A description of any assumptions or corrections, such as tests of normality and adjustment for multiple comparisons |
| ☒ | ☐ | A full description of the statistical parameters including central tendency (e.g. means) or other basic estimates (e.g. regression coefficient) AND variation (e.g. standard deviation) or associated estimates of uncertainty (e.g. confidence intervals) |
| ☒ | ☐ | For null hypothesis testing, the test statistic (e.g. *F*, *t*, *r*) with confidence intervals, effect sizes, degrees of freedom and *P* value noted *Give P values as exact values whenever suitable.* |
| ☒ | ☐ | For Bayesian analysis, information on the choice of priors and Markov chain Monte Carlo settings |
| ☐ | ☒ | For hierarchical and complex designs, identification of the appropriate level for tests and full reporting of outcomes |
| ☒ | ☐ | Estimates of effect sizes (e.g. Cohen's *d*, Pearson's *r*), indicating how they were calculated |

*Our web collection on statistics for biologists contains articles on many of the points above.*

## Software and code

Policy information about availability of computer code

| Data collection | *Provide a description of all commercial, open source and custom code used to collect the data in this study, specifying the version used OR state that no software was used.* |
|---|---|
| Data analysis | We applied commercial software Matlab and Abaqus for computations. |

For manuscripts utilizing custom algorithms or software that are central to the research but not yet described in published literature, software must be made available to editors and reviewers. We strongly encourage code deposition in a community repository (e.g. GitHub). See the Nature Research guidelines for submitting code & software for further information.

## Data

Policy information about availability of data

All manuscripts must include a data availability statement. This statement should provide the following information, where applicable:
- Accession codes, unique identifiers, or web links for publicly available datasets
- A list of figures that have associated raw data
- A description of any restrictions on data availability

The data used in this study are available from the corresponding authors upon reasonable request. Source data are provided with this paper.

# Field-specific reporting

Please select the one below that is the best fit for your research. If you are not sure, read the appropriate sections before making your selection.

☐ Life sciences      ☐ Behavioural & social sciences      ☐ Ecological, evolutionary & environmental sciences

For a reference copy of the document with all sections, see nature.com/documents/nr-reporting-summary-flat.pdf

# Life sciences study design

All studies must disclose on these points even when the disclosure is negative.

| | |
|---|---|
| Sample size | *Describe how sample size was determined, detailing any statistical methods used to predetermine sample size OR if no sample-size calculation was performed, describe how sample sizes were chosen and provide a rationale for why these sample sizes are sufficient.* |
| Data exclusions | *Describe any data exclusions. If no data were excluded from the analyses, state so OR if data were excluded, describe the exclusions and the rationale behind them, indicating whether exclusion criteria were pre-established.* |
| Replication | *Describe the measures taken to verify the reproducibility of the experimental findings. If all attempts at replication were successful, confirm this OR if there are any findings that were not replicated or cannot be reproduced, note this and describe why.* |
| Randomization | *Describe how samples/organisms/participants were allocated into experimental groups. If allocation was not random, describe how covariates were controlled OR if this is not relevant to your study, explain why.* |
| Blinding | *Describe whether the investigators were blinded to group allocation during data collection and/or analysis. If blinding was not possible, describe why OR explain why blinding was not relevant to your study.* |

# Behavioural & social sciences study design

All studies must disclose on these points even when the disclosure is negative.

| | |
|---|---|
| Study description | *Briefly describe the study type including whether data are quantitative, qualitative, or mixed-methods (e.g. qualitative cross-sectional, quantitative experimental, mixed-methods case study).* |
| Research sample | *State the research sample (e.g. Harvard university undergraduates, villagers in rural India) and provide relevant demographic information (e.g. age, sex) and indicate whether the sample is representative. Provide a rationale for the study sample chosen. For studies involving existing datasets, please describe the dataset and source.* |
| Sampling strategy | *Describe the sampling procedure (e.g. random, snowball, stratified, convenience). Describe the statistical methods that were used to predetermine sample size OR if no sample-size calculation was performed, describe how sample sizes were chosen and provide a rationale for why these sample sizes are sufficient. For qualitative data, please indicate whether data saturation was considered, and what criteria were used to decide that no further sampling was needed.* |
| Data collection | *Provide details about the data collection procedure, including the instruments or devices used to record the data (e.g. pen and paper, computer, eye tracker, video or audio equipment) whether anyone was present besides the participant(s) and the researcher, and whether the researcher was blind to experimental condition and/or the study hypothesis during data collection.* |
| Timing | *Indicate the start and stop dates of data collection. If there is a gap between collection periods, state the dates for each sample cohort.* |
| Data exclusions | *If no data were excluded from the analyses, state so OR if data were excluded, provide the exact number of exclusions and the rationale behind them, indicating whether exclusion criteria were pre-established.* |
| Non-participation | *State how many participants dropped out/declined participation and the reason(s) given OR provide response rate OR state that no participants dropped out/declined participation.* |
| Randomization | *If participants were not allocated into experimental groups, state so OR describe how participants were allocated to groups, and if allocation was not random, describe how covariates were controlled.* |

# Ecological, evolutionary & environmental sciences study design

All studies must disclose on these points even when the disclosure is negative.

| | |
|---|---|
| Study description | *Briefly describe the study. For quantitative data include treatment factors and interactions, design structure (e.g. factorial, nested, hierarchical), nature and number of experimental units and replicates.* |
| Research sample | *Describe the research sample (e.g. a group of tagged Passer domesticus, all Stenocereus thurberi within Organ Pipe Cactus National* |

| Research sample | *Monument), and provide a rationale for the sample choice. When relevant, describe the organism taxa, source, sex, age range and any manipulations. State what population the sample is meant to represent when applicable. For studies involving existing datasets, describe the data and its source.* |
|---|---|
| Sampling strategy | *Note the sampling procedure. Describe the statistical methods that were used to predetermine sample size OR if no sample-size calculation was performed, describe how sample sizes were chosen and provide a rationale for why these sample sizes are sufficient.* |
| Data collection | *Describe the data collection procedure, including who recorded the data and how.* |
| Timing and spatial scale | *Indicate the start and stop dates of data collection, noting the frequency and periodicity of sampling and providing a rationale for these choices. If there is a gap between collection periods, state the dates for each sample cohort. Specify the spatial scale from which the data are taken* |
| Data exclusions | *If no data were excluded from the analyses, state so OR if data were excluded, describe the exclusions and the rationale behind them, indicating whether exclusion criteria were pre-established.* |
| Reproducibility | *Describe the measures taken to verify the reproducibility of experimental findings. For each experiment, note whether any attempts to repeat the experiment failed OR state that all attempts to repeat the experiment were successful.* |
| Randomization | *Describe how samples/organisms/participants were allocated into groups. If allocation was not random, describe how covariates were controlled. If this is not relevant to your study, explain why.* |
| Blinding | *Describe the extent of blinding used during data acquisition and analysis. If blinding was not possible, describe why OR explain why blinding was not relevant to your study.* |

Did the study involve field work? ☐ Yes ☒ No

# Reporting for specific materials, systems and methods

We require information from authors about some types of materials, experimental systems and methods used in many studies. Here, indicate whether each material, system or method listed is relevant to your study. If you are not sure if a list item applies to your research, read the appropriate section before selecting a response.

## Materials & experimental systems

| n/a | Involved in the study |
|---|---|
| ☒ | ☐ Antibodies |
| ☒ | ☐ Eukaryotic cell lines |
| ☒ | ☐ Palaeontology and archaeology |
| ☒ | ☐ Animals and other organisms |
| ☒ | ☐ Human research participants |
| ☒ | ☐ Clinical data |
| ☒ | ☐ Dual use research of concern |

## Methods

| n/a | Involved in the study |
|---|---|
| ☒ | ☐ ChIP-seq |
| ☒ | ☐ Flow cytometry |
| ☒ | ☐ MRI-based neuroimaging |

