## [Peer review file · Nature Computational Science]

Peer Review Information

Journal: Nature Computational Science

Manuscript Title: Chiral Topographic Instability in Shrinking Spheres

Corresponding author name(s): Fan Xu, Xiqiao Feng

Reviewer Comments & Decisions:

Decision Letter, initial version:

Date: 16th June 22 16:07:19

Last Sent: 16th June 22 16:07:19

Triggered By: Jie Pan

From: jie.pan@us.nature.com

To: fanxu@fudan.edu.cn

BCC: jie.pan@us.nature.com

Subject: Decision on Nature Computational Science manuscript NATCOMPUTSCI-22-0393

Message: ** Please ensure you delete the link to your author homepage in this e-mail if you wish to forward it to your co-authors. **

Dear Professor Xu,

Your manuscript "Chiral Topographic Instability in Shrinking Spheres" has now been seen by 3 referees, whose comments are appended below. You will see that while they find your work of interest, they have raised points that need to be addressed before we can make a decision on publication.

The referees' reports seem to be quite clear. Naturally, we will need you to address all of the points raised.

While we ask you to address all of the points raised, the following points need to be substantially worked on:

- Please add more discussions about relevant literatures as suggested by referees.
- Please include more details about model and methodology. You could use Methods section for these details, and we do not have a word limit for the Methods section.
- Please add comparisons between the buckling simulation and experimental

measurement.

- Please clarify technical points as suggested by our referees.
- Please better highlight about the advance of your method over the existing methods, such as FEM.
- Per our formatting requirements, please split your discussions into Introduction, Results, Discussion (where limitations shall be discussed), and Methods.

Please use the following link to submit your revised manuscript and a point-by-point response to the referees' comments (which should be in a separate document to any cover letter):

[REDACTED]

** This url links to your confidential homepage and associated information about manuscripts you may have submitted or be reviewing for us. If you wish to forward this e-mail to co-authors, please delete this link to your homepage first. **

To aid in the review process, we would appreciate it if you could also provide a copy of your manuscript files that indicates your revisions by making use of Track Changes or similar mark-up tools. Please also ensure that all correspondence is marked with your Nature Computational Science reference number in the subject line.

In addition, please make sure to upload a Word Document or LaTeX version of your text, to assist us in the editorial stage.

To improve transparency in authorship, we request that all authors identified as 'corresponding author' on published papers create and link their Open Researcher and Contributor Identifier (ORCID) with their account on the Manuscript Tracking System (MTS), prior to acceptance. ORCID helps the scientific community achieve unambiguous attribution of all scholarly contributions. You can create and link your ORCID from the home page of the MTS by clicking on 'Modify my Springer Nature account'. For more information please visit www.springernature.com/orcid.

We hope to receive your revised paper within three weeks. If you cannot send it within this time, please let us know.

Best regards,

Jie Pan, Ph.D.
Associate Editor
Nature Computational Science

Reviewers comments:

Reviewer #1 (Remarks to the Author):

This work reports the chiral wrinkling of passion fruits upon dehydrate, which evolves from a buckyball pattern to a chiral mode, and further forms a chiral topology network. To understand these phenomena, the authors model the fruit as an elastic shell with a soft core. The elastic strain energy of shell is composed of bending and membrane energy. The elastic behavior of the core is simplified as a Winkler foundation. The buckling of core-shell sphere upon shrinkage is resembled to the instability of a spherical shell under hydrostatic pressure. Combined with Kolter's theory, divergence theorem, and assumed displacement field, the critical condition of the buckling onset is established. To provide further insights, the Y shape wrinkling of the buckyball pattern is modeled as buckling of a bilayer under compression. By introducing velocity-dependent damping and inertia terms, simulations with different material properties and geometries are conducted to trace the post buckling process. Guided by these theory analyses, the authors also design a buckling device, realizing picking up and releasing of objects with different geometry.

This paper proposes a new theory and scaling law with some fundament understanding of chiral buckling behaviors and inspiring engineering design based on them. The manuscript is well prepared with a smooth and clear logical flow. Therefore, the paper is recommended for publication with only some minor questions:

1. Page 2, Paragraph 2. It seems there is not much background literature review on the related topic. What is the state of art of similar works in the field?
2. Page 3, first sentence, "Young's modulus and Poisson's ratio of the shell are respectively denoted by E_f and ν_f ". In real scenario, the modulus of a fruit shell may change a lot upon dehydration. How to justify the small deformation or reconcile the large deformation during this process?
3. Page 7, Figure 3, is it possible for the author to provide some comparison between the buckling simulation and experimental measurement?
4. Page 8, first sentence, "We slowly extracted air from the sample to control the pressure". What is the air velocity and how large is the pressure?

Reviewer #2 (Remarks to the Author):

With the purpose of driving the design of grasping devices, the authors successfully analyze the instability patterns arising in shrinking spheres. More specifically, the chiral patterns disclosed by the authors are a second bifurcation stage following the bucklyball pattern (previously investigated by other authors) and are detected through a blend of analytical and numerical tools. The research is complemented by experimental tests of shrinking and punching confirming the theoretically predicted chiral patterns, as well as the experimental grasping validation with different objects.

The research is fair and interesting, therefore acceptance for publication in Nature Computational Science is recommended, subject to addressing the following major remark.

Major remark:

The mechanical model is not completely described in the main text. Moreover, since a spherical geometry is investigated it is not clear why a cartesian reference system is considered.

Minor remarks:

- The expression for the core stiffness K_s appearing after eqn (2) should be explained or at least enhanced by adding a reference;
- In Section I of supplementary material, "an upper layer h_f " should be "an upper layer of width h_f " and similarly for the lower layer.

Reviewer #3 (Remarks to the Author):

The article presents a mixture of an analytical and numerical study of chiral pattern formation in shrinking spheres. The analytical model is based on linear principles where buckling eigenvalues and eigenvectors are obtained. The numerical modelling is conducted in the commercial code ABAQUS and is based geometrically nonlinear principles. While the submission is well presented, there are some issues with the claims that are made that need some attention.

1) "Buckyball" vs hexagonal deformation pattern. From the numerical results presented in Figure 1 and the movies supplied, it was difficult to observe whether the "buckyball" mode was formed with its characteristic pentagons and hexagons; it appears the critical mode shown is purely hexagonal and this transforms into the chiral mode. Perhaps the authors could clarify this point.

2) The equilibrium paths shown in Figure 3 are found using ABAQUS. Hence, the reviewer would not expect true branching points to be detected when secondary instabilities are subsequently triggered. This is because the arc-length method within ABAQUS does not have the necessary "generalised path-following" algorithms that can detect such branch points in other codes (see for instance the excellent work by Groh et al from the University of Bristol on such FE methods). Using the dynamic relaxation method can get around this issue but is there any way of guaranteeing that the solution is on the path that the actual structure would follow? Or was this assumed to be correct given that it resembled the observed results in the physical experiments? Again, perhaps the authors could comment on this matter.

Author Rebuttal to Initial comments

Response to
Reviewers

Title	Chiral Topographic Instability in Shrinking Spheres
Authors	Fan Xu, Yangchao Huang, Shichen Zhao, Xi-Qiao Feng
Journal	Nature Computational Science
Manuscript No.	NATCOMPUTSCI-22-0393

Reviewer #1

- **Comment (o)**

This work reports the chiral wrinkling of passion fruits upon dehydrate, which evolves from a buckyball pattern to a chiral mode, and further forms a chiral topology network. To understand these phenomena, the authors model the fruit as an elastic shell with a soft core. The elastic strain energy of shell is composed of bending and membrane energy. The elastic behavior of the core is simplified as a Winkler foundation. The buckling of core-shell sphere upon shrinkage is resembled to the instability of a spherical shell under hydrostatic pressure. Combined with Kolter's theory, divergence theorem, and assumed displacement field, the critical condition of the buckling onset is established. To provide further insights, the Y shape wrinkling of the buckyball pattern is modeled as buckling of a bilayer under compression. By introducing velocity-dependent damping and inertia terms, simulations with different material properties and geometries are conducted to trace the post buckling process. Guided by these theory analyses, the authors also design a buckling device, realizing picking up and releasing of objects with different geometry.

This paper proposes a new theory and scaling law with some fundament understanding of chiral buckling behaviors and inspiring engineering design based on them. The manuscript is well prepared with a smooth and clear logical flow. Therefore, the paper is recommended for publication with only some minor questions.

- ✓ **Response (o)**

We thank the reviewer for the insightful, encouraging comments and very comprehensive summary of the work. We also appreciate that the significance and broad interests from fundamental understanding to engineering design application of this work have been highlighted by the referee.

We thank the reviewer for his/her very careful reading and kind suggestions to help us improve the manuscript. All the concerns raised by the reviewer are addressed point-by-point in the following.

- **Comment (1)**

Page 2, Paragraph 2. It seems there is not much background literature review on the related topic. What is the state of art of similar works in the field?

- ✓ **Response (1)**

We thank the reviewer for this kind suggestion. In paragraph 1, we cited a number of works on the related topic and discussed their general state of the art in biological functions and practical applications via harnessing morphology patterns. As suggested by the reviewer, we have now added some discussions on the more specific background on morphological pattern formation in spherical core-shells in the beginning of paragraph 2, page 2 for clarification.

“Prior works [Cao et al., 2008; Li et al., 2011; Bred & Crosby, 2013; Stoop et al., 2015; Xu et al., 2020] on morphological pattern formation in stressed spherical core-shells, a typical structure omnipresent in nature and industrial technologies, demonstrated a variety of intriguing topographies such as dimple, buckyball and labyrinth modes. Here, we report a novel chiral instability topography in core-shell spheres.”

G. Cao, X. Chen, C. Li, A. Ji, and Z. Cao, Self-assembled triangular and labyrinth buckling patterns of thin films on spherical substrates, *Phys. Rev. Lett.* 100, 036102 (2008).

D. Breid and A.J. Crosby, Curvature-controlled wrinkle morphologies, *Soft Matter* 9, 3624 (2013).

F. Xu, S. Zhao, C. Lu, and M. Potier-Ferry, Pattern selection in core-shell spheres. *J. Mech. Phys. Solids* 137, 103892 (2020).

- **Comment (2)**

Page 3, first sentence, “Young’s modulus and Poisson’s ratio of the shell are respectively denoted by E_f and ν_f ”. In real scenario, the modulus of a fruit shell may change a lot upon dehydration. How to justify the small deformation or reconcile the large deformation during this process?

✓ **Response (2)**

This is a good point and we agree on this comment. During the natural dehydration process of passion fruits, the moduli of both surface layer and soft core may become larger (stiffer), but we observed that the wrinkling wavelength in experiments remains almost unchanged, and this critical wavelength ℓ_{cr}

has some inherent (yet implicit) relation with the modulus ratio E_s/E_f (see Eq. (5) in the main text). Therefore, we suspect that the modulus ratio E_s/E_f remains relatively constant upon dehydration. We defined a key dimensionless parameter $C_s = (E_s/E_f) (R/h_f)^{3/2}$ that characterizes the stiffness ratio of core-shells and geometric curvature to classify pattern selection in the original manuscript.

For numerical predictions of the entire postbuckling evolution, as mentioned in Methods section in the original manuscript, “since the deformation of core-shell spheres can be large (up to 30% shrinking strain), we applied widely used hyperelastic neo-Hookean (nHk) constitutive law for both surface layer and soft core, while more sophisticated hyperelastic constitutions such as Mooney-Rivlin (MR) were also examined, which showed insignificant quantitative differences that have not changed the substantial nonlinear mechanism of the instability problem.” For theoretical predictions of critical buckling conditions and dimensional analysis on chiral instability, we considered geometric nonlinearity and limit ourselves within linear elastic material (Hooke’s law) for simplicity. In fact, it is normally impossible to analytically deal with both material and geometric nonlinearity together, especially for large, inhomogeneous deformations of core-shells considered here, while such simplification is sufficient for the critical buckling analysis and scaling law considered in this work (see Fig. 2 in the main text).

We have added the following explanation in the main text for clarification.

“During the natural dehydration process of passion fruits, the moduli of both surface layer and soft core may become larger (stiffer), but we observed that the wrinkling wavelength in experiments remains almost unchanged, and this critical wavelength ℓ_{cr} has some inherent (yet implicit) relation

with the modulus ratio E_s/E_f (see Eq. (5)). Therefore, it is reasonable to suppose that the modulus ratio E_s/E_f remains relatively constant upon dehydration.”

• **Comment (3)**

Page 7, Figure 3, is it possible for the author to provide some comparison between the buckling simulation and experimental measurement?

✓ **Response (3)**

Thanks for this kind suggestion. We have added qualitative comparison of diverse characteristic wrinkling topographies between computations and experiments in Fig. 3. In fact, it is really difficult to carry out more quantitative comparison on bifurcation evolution, since the true shrinking strain in experiments is hard to be precisely measured.

FIG. 3

- **Comment (4)**

Page 8, first sentence, “We slowly extracted air from the sample to control the pressure”. What is the air velocity and how large is the pressure?

- ✓ **Response (4)**

In our experiments, the air velocity is about 2 ml/s and the pressure is around 10 KPa when the hexagonal-to-chiral mode transition occurs. We have added the information in the main text.

“We slowly extracted air (~2 ml/s) from the sample to control the pressure (~10 KPa) so that a state of homogeneous compression can be perfectly achieved.”

Reviewer #2

- **Comment (0)**

With the purpose of driving the design of grasping devices, the authors successfully analyze the instability patterns arising in shrinking spheres. More specifically, the chiral patterns disclosed by the authors are a second bifurcation stage following the bucklyball pattern (previously investigated by other authors) and are detected through a blend of analytical and numerical tools. The research is complemented by experimental tests of shrinking and punching confirming the theoretically predicted chiral patterns, as well as the experimental grasping validation with different objects.

The research is fair and interesting, therefore acceptance for publication in Nature Computational Science is recommended, subject to addressing the following major remark.

- ✓ **Response (0)**

We thank the reviewer for the comprehensive summary and insightful comments of the work. We also thank the reviewer for his/her very careful reading and kind suggestions to help us improve the manuscript. All the concerns raised by the reviewer are addressed point-by-point in the following.

- **Comment (1)**

Major remark:

The mechanical model is not completely described in the main text. Moreover, since a spherical geometry is investigated it is not clear why a cartesian reference system is considered.

✓ **Response (1)**

Thanks for this comment. We did not describe the details of the model in the main text. The core-shell spherical system was considered to be three-dimensional and the geometry was given in Fig. S1 in Supplementary Material. Theoretical treatment of this system can be Cartesian in a tangent plane or curvilinear and orthogonal, both of which are equivalent within the shallow shell theory (van der Heijden, 2009) and can only describe a small part of spherical shell, but it is sufficient here for theoretical analyses. We have added some explanations in the main text for clarification.

“Theoretical treatment of the core-shell system can be Cartesian in a tangent plane or curvilinear and orthogonal, both of which are equivalent within the shallow shell theory (van der Heijden, 2009). This framework can only describe a part of spherical geometry (see Supplementary Fig. S1), but it is sufficient here for theoretical analyses.”

• **Comment (2)**

Minor remarks:

- The expression for the core stiffness K_s appearing after eqn (2) should be explained or at least enhanced by adding a reference;

✓ **Response (2)**

Thanks for this comment. This expression suggests a Winkler-type foundation that the stiffness is dependent on the half buckling wavelength $K_s \sim E_s/\ell_{cr}$ (Biot, JAM, 1937), and well accounts for the initial curvature of the system, satisfying $\pi/\ell_{cr} = \sqrt{p^2 + q^2}/R$ (Zhao et al., JMPS, 2014; Xu et al., JMPS, 2020). We have added relevant references in the main text.

“...in which $K_s = \overline{E_s} \sqrt{p^2 + q^2} / 2R$ denotes the stiffness of the core [Zhao et al., JMPS, 2014; Xu et al., JMPS, 2020]”

Y. Zhao, Y. Cao, X. Feng, and K. Ma, Axial compression-induced wrinkles on a core-shell soft cylinder: Theoretical analysis, simulations and experiments, *J. Mech. Phys. Solids* 73, 212 (2014).

F. Xu, S. Zhao, C. Lu, and M. Potier-Ferry, Pattern selection in core-shell spheres, *J. Mech. Phys. Solids*, 137, 103892 (2020).

- **Comment (3)**

- In Section I of supplementary material, "an upper layer h_f " should be "an upper layer of width h_f " and similarly for the lower layer.

- ✓ **Response (3)**

Thanks for this kind suggestion. We have made the corresponding modification in Supplementary Material.

“...an upper layer of width h_f and a lower layer of width h_s .”

Reviewer #3

- **Comment (0)**

The article presents a mixture of an analytical and numerical study of chiral pattern formation in shrinking spheres. The analytical model is based on linear principles where buckling eigenvalues and eigenvectors are obtained. The numerical modelling is conducted in the commercial code ABAQUS and is based geometrically nonlinear principles. While the submission is well presented, there are some issues with the claims that are made that need some attention.

- ✓ **Response (0)**

We thank the reviewer for the valuable comments and kind suggestions to help us improve the manuscript. All the concerns raised by the reviewer are addressed point-by-point in the following.

- **Comment (1)**

"Buckyball" vs hexagonal deformation pattern. From the numerical results presented in Figure 1 and the movies supplied, it was difficult to observe whether the "buckyball" mode was formed with its characteristic pentagons and hexagons; it appears the critical mode shown is purely hexagonal and

this transforms into the chiral mode. Perhaps the authors could clarify this point.

✓ **Response (1)**

We thank the reviewer for this comment. To more clearly demonstrate the “buckyball” mode (a mixture of periodic hexagons and pentagons), we have highlighted the contour profile of buckyball mode in Fig. 1 (f), where a mixture of hexagons and pentagons can be observed, but hexagons are mostly prevailing.

We provided relevant interpretation on the buckyball pattern in the previous version of manuscript.

“Note that although both natural and numerical observations (see Fig. 1(b) and (f)) show that the buckyball pattern consisting of hexagons and pentagons covers the whole sphere (non-developable surface), hexagon is the prevailing buckling mode in core-shell spheres.”

FIG. 1

• **Comment (2)**

The equilibrium paths shown in Figure 3 are found using ABAQUS. Hence, the reviewer would not expect true branching points to be detected when secondary instabilities are subsequently triggered.

This is because the arc-length method within ABAQUS does not have the necessary "generalised path-following" algorithms that can detect such branch points in other codes (see for instance the excellent work by Groh et al from the University of Bristol on such FE methods). Using the dynamic relaxation method can get around this issue but is there any way of guaranteeing that the solution is on the path that the actual structure would follow? Or was this assumed to be correct given that it resembled the observed results in the physical experiments? Again, perhaps the authors could comment on this matter.

✓ **Response (2)**

We thank the reviewer for raising this good point. Although some path-following arc-length methods such as Riks and Asymptotic Numerical Method (Damil & Potier-Ferry, *IJES*, 28, 943-957, 1990; Xu & Potier-Ferry, *JMPS*, 94, 68-87, 2016) can predict most of conventional nonlinear responses in the presence of both subcritical and supercritical bifurcations as well as hysteresis loops, the convergence is not always ensured. For instance, when solving a difficult buckling problem of a shearing thin membrane, Wong and Pellegrino (*J. Mech. Mater. Struct.* 1, 63-95, 2006) concluded that "all attempts to use the arc-length solution method in Abaqus (Riks) were unsuccessful". With the same path-following algorithm, we encountered the same convergence difficulties in the present case of extremely deformed core-shell spheres upon excessive shrinking, which thus motivated us to pursue an alternative technique to extend the range of numerical methods, namely a pseudodynamic algorithm.

In the previous version of manuscript, we provided the relevant discussions on numerical algorithms.

"The main challenge lies in the resolution of nonlinear equations, since a large number of solution branches in the postbuckling regime are possible and can be connected via multiple bifurcations. Moreover, for instabilities that are extremely localized, e.g., ridge network in Fig. 1(c)-(d), there may exist a local transfer of strain energy from one part of the model to the neighboring parts, and global resolution methods may encounter difficulties in convergence. To solve this difficulty, we implemented a pseudodynamic algorithm through introducing velocity-dependent damping and inertial terms, which can be naturally viewed as a perturbation to allow the calculation to pass through the unstable transitions and to trigger chiral symmetry breaking (see Methods section)."

“When the model is stable (quasi-static), viscous energy dissipation remains very small such that the artificial damping does not perturb the solution significantly. When the system goes dynamically unstable, however, nodal velocities increase and then part of the strain energy released is dissipated by the damping.”

To ensure the solution is on the path that the actual structure would follow, we carried out comparisons between the pseudodynamic algorithm and path-following method (Riks) when the load remains relatively not large (around first bifurcation and its postbuckling regime). The path evolutions of both methods are consistent (see examples in Supplementary Material Ref. [2] Xu et al., *JMPS*, 2020). Besides, qualitative comparisons of diverse characteristic wrinkling topographies between computations and experiments (natural dehydration of passion fruits) are now provided in Fig. 3, which partially verifies the accuracy and reliability of the numerical predictions.

We thank the reviewer for drawing our attention to the excellent work by Groh et al. (*CMAME*, 331, 394-426, 2018). We have cited it in Methods section in the main text.

“The main challenge is the resolution of nonlinear equations, since a large number of postbuckling solution branches are possible and can be connected via multiple bifurcations [Groh et al., 2018; Xu et al., 2020].”

R.M.J. Groh, D. Avitabile, and A. Pirrera, Generalised path-following for well-behaved nonlinear structures, *Comput. Methods. Appl. Mech. Eng.* 331, 394 (2018).

The “generalised” path-following method proposed by Groh et al. was shown to be a powerful algorithm that can continuously capture multiple bifurcation points and associated branching paths in multi-stable thin-walled structures such as fibre-reinforced composite laminates where geometric nonlinearity is the major concern. The method costs no extra computational expense due to the use of the branch-connecting approach when additional equilibrium paths exist. Any parameters that involve nonlinear problems, such as geometric dimensions, constitutive properties and loads, can be followed on a multi-dimensional solution manifold. We are willing to see its potential extension to the hyperelastic composite structures such as soft core-shells involving both strong material and geometric nonlinearities in future work.

Decision Letter, first revision:

Date: 1st August 22 05:17:11
Last Sent: 1st August 22 05:17:11
Triggered By: Jie Pan
From: jie.pan@us.nature.com
To: fanxu@fudan.edu.cn
CC: computacionalscience@nature.com
BCC: jie.pan@us.nature.com
Subject: AIP Decision on Manuscript NATCOMPUTSCI-22-0393A
Message: Our ref: NATCOMPUTSCI-22-0393A

1st August 2022

Dear Dr. Xu,

Thank you for submitting your revised manuscript "Chiral Topographic Instability in Shrinking Spheres" (NATCOMPUTSCI-22-0393A). It has now been seen by the original referees and their comments are below. The reviewers find that the paper has improved in revision, and therefore we'll be happy in principle to publish it in Nature Computational Science, pending minor revisions to satisfy the referees' final requests and to comply with our editorial and formatting guidelines.

TRANSPARENT PEER REVIEW

Nature Computational Science offers a transparent peer review option for original research manuscripts. We encourage increased transparency in peer review by publishing the reviewer comments, author rebuttal letters and editorial decision letters if the authors agree. Such peer review material is made available as a supplementary peer review file. **Please state in the cover letter 'I wish to participate in transparent peer review' if you want to opt in, or 'I do not wish to participate in transparent peer review' if you don't.** Failure to state your preference will result in delays in accepting your manuscript for publication.

Please note: we allow redactions to authors' rebuttal and reviewer comments in the interest of confidentiality. If you are concerned about the release of confidential data, please let us know specifically what information you would like to have removed.

Please note that we cannot incorporate redactions for any other reasons. Reviewer names will be published in the peer review files if the reviewer signed the comments to authors, or if reviewers explicitly agree to release their name. For more information, please refer to our [FAQ page](https://www.nature.com/documents/nr-transparent-peer-review.pdf).

Thank you again for your interest in Nature Computational Science Please do not

hesitate to contact me if you have any questions.

Sincerely,

Jie Pan, Ph.D.
Associate Editor
Nature Computational Science

ORCID

Reviewer #1 (Remarks to the Author):

The author has successfully addressed my previous comments. The paper is now recommended for publication.

Reviewer #2 (Remarks to the Author):

The authors have properly addressed the comments raised by the reviewer. Acceptance for publication is recommended.

I reviewed the codes material and:

- I can confirm that instructions file is present, which contains clear explanations about how to use the code files
- The codes in Matlab are perfectly running and reproduce the results presented in the paper
- The cae file is sensitive to the Abaqus version, so I would suggest to the authors to include an input file too for users compatibility. Anyway, I have been able to open it in Abaqus 2021 and the conversion was successful, then just tried to run one of the many jobs modeled within the cae file and it runs. However, I did not complete the simulation because it was taking too long for my device.

Reviewer #3 (Remarks to the Author):

Thank you for addressing the comments. I now recommend publication and congratulate the authors for their excellent work.

Final Decision Letter:**Date:** 9th September 22 03:10:23**Last Sent:** 9th September 22 03:10:23**Triggered By:** Jie Pan**From:** jie.pan@us.nature.com**To:** fanxu@fudan.edu.cn**Subject:** Decision on Nature Computational Science manuscript NATCOMPUTSCI-22-0393B**Message:** Dear Professor Xu,

We are pleased to inform you that your Article "Chiral Topographic Instability in Shrinking Spheres" has now been accepted for publication in Nature Computational Science.

Please note that *Nature Computational Science* is a Transformative Journal (TJ). Authors may publish their research with us through the traditional subscription access route or make their paper immediately open access through payment of an article-processing charge (APC). Authors will not be required to make a final decision about access to their article until it has been accepted. [Find out more about Transformative Journals](https://www.springernature.com/gp/open-research/transformative-journals)

Acceptance of your manuscript is conditional on all authors' agreement with our publication policies (see <https://www.nature.com/natcomputsci/for-authors>). In

particular your manuscript must not be published elsewhere and there must be no announcement of the work to any media outlet until the publication date (the day on which it is uploaded onto our web site).

Before your manuscript is typeset, we will edit the text to ensure it is intelligible to our wide readership and conforms to house style. We look particularly carefully at the titles of all papers to ensure that they are relatively brief and understandable.

Once your manuscript is typeset and you have completed the appropriate grant of rights, you will receive a link to your electronic proof via email with a request to make any corrections within 48 hours. If, when you receive your proof, you cannot meet this deadline, please inform us at rjsproduction@springernature.com immediately.

If you have queries at any point during the production process then please contact the production team at rjsproduction@springernature.com. Once your paper has been scheduled for online publication, the Nature press office will be in touch to confirm the details.

Content is published online weekly on Mondays and Thursdays, and the embargo is set at 16:00 London time (GMT)/11:00 am US Eastern time (EST) on the day of publication. If you need to know the exact publication date or when the news embargo will be lifted, please contact our press office after you have submitted your proof corrections. Now is the time to inform your Public Relations or Press Office about your paper, as they might be interested in promoting its publication. This will allow them time to prepare an accurate and satisfactory press release. Include your manuscript tracking number NATCOMPUTSCI-22-0393B and the name of the journal, which they will need when they contact our office.

About one week before your paper is published online, we shall be distributing a press release to news organizations worldwide, which may include details of your work. We are happy for your institution or funding agency to prepare its own press release, but it must mention the embargo date and Nature Computational Science. Our Press Office will contact you closer to the time of publication, but if you or your Press Office have any inquiries in the meantime, please contact press@nature.com.

We welcome the submission of potential cover material (including a short caption of around 40 words) related to your manuscript; suggestions should be sent to Nature Computational Science as electronic files (the image should be 300 dpi at 210 x 297 mm in either TIFF or JPEG format). We also welcome suggestions for the Hero Image, which appears at the top of our <http://www.nature.com/natcomputsci> home page; these should be 72 dpi at 1400 x 400 pixels in JPEG format. Please note that such pictures should be selected more for their aesthetic appeal than for their scientific content, and that colour images work better than black and white or grayscale images. Please do not try to design a cover with the Nature Computational Science logo etc., and please do not

submit composites of images related to your work. I am sure you will understand that we cannot make any promise as to whether any of your suggestions might be selected for the cover of the journal.

Best regards,

Jie Pan, Ph.D.
Associate Editor
Nature Computational Science

P.S. Click on the following link if you would like to recommend Nature Computational Science to your librarian: https://www.springernature.com/gp/librarians/recommend-to-your-library

** Visit the Springer Nature Editorial and Publishing website at www.springernature.com/editorial-and-publishing-jobs for more information about our career opportunities. If you have any questions please click here. **